# Mechanism of the Conformational Change of the Protein Methyltransferase SMYD3: A Molecular Dynamics Simulation Study

**DOI:** 10.3390/ijms22137185

**Published:** 2021-07-02

**Authors:** Jixue Sun, Zibin Li, Na Yang

**Affiliations:** State Key Laboratory of Medicinal Chemical Biology, College of Pharmacy and Key Laboratory of Medical Data Analysis and Statistical Research of Tianjin, Nankai University, Tianjin 300353, China; JixueSun@nankai.edu.cn (J.S.); 9820200006@nankai.edu.cn (Z.L.)

**Keywords:** methyltransferase, conformational change, MD simulation, molecular mechanism, binding free energy

## Abstract

SMYD3 is a SET-domain-containing methyltransferase that catalyzes the transfer of methyl groups onto lysine residues of substrate proteins. Methylation of MAP3K2 by SMYD3 has been implicated in Ras-driven tumorigenesis, which makes SMYD3 a potential target for cancer therapy. Of all SMYD family proteins, SMYD3 adopt a closed conformation in a crystal structure. Several studies have suggested that the conformational changes between the open and closed forms may regulate the catalytic activity of SMYD3. In this work, we carried out extensive molecular dynamics simulations on a series of complexes with a total of 21 μs sampling to investigate the conformational changes of SMYD3 and unveil the molecular mechanisms. Based on the C-terminal domain movements, the simulated models could be depicted in three different conformational states: the closed, intermediate and open states. Only in the case that both the methyl donor binding pocket and the target lysine-binding channel had bound species did the simulations show SMYD3 maintaining its conformation in the closed state, indicative of a synergetic effect of the cofactors and target lysine on regulating the conformational change of SMYD3. In addition, we performed analyses in terms of structure and energy to shed light on how the two regions might regulate the C-terminal domain movement. This mechanistic study provided insights into the relationship between the conformational change and the methyltransferase activity of SMYD3. The more complete understanding of the conformational dynamics developed here together with further work may lay a foundation for the rational drug design of SMYD3 inhibitors.

## 1. Introduction

SMYD3 is a member of the family of SMYD (SET- and MYND-domain-containing) lysine methyltransferases, which catalyze the methylations of lysines of various histone and non-histone targets [1,2]. The reaction catalyzed by SMYD3 employs *S*-adenosyl-l-methionine (SAM) as the methyl donor to methylate the Nε atom of a lysine residue in the target protein substrate and leaves *S*-adenosyl-l-homocysteine (SAH) as the cofactor product. The reported histone targets of SMYD3 include histones H3K4, H4K5, H4K20 and histone variant H2A.Z.1K101 [3,4,5,6], highlighting its key role in affecting transcriptional regulation [7,8,9]. Additionally, several important kinases, including vascular endothelial growth factor receptor 1 (VEGFR1, K831), MAP3 kinase kinase (MAP3K2, K260), v-Akt murine thymoma viral oncogene homolog 1 (AKT1, K14) and human epidermal growth factor receptor 2 (HER2, K175), have been reported to be nonhistone substrates of SMYD3 [10,11,12,13].

As its methyltransferase activity on nonhistone targets involves tumorigenesis, SMYD3 has drawn attention for its role in cancer progression and invasion [14,15]. It has been implicated in a variety of cancers including breast, liver, lung, and pancreatic cancers [16,17,18,19,20]. Methylation of K831 on VEGFR1 by SMYD3 increases kinase activity through ligand-dependent autophosphorylation in the cell to promote breast and pancreatic carcinomas [21,22,23,24]. In lung and pancreatic cancers, SMYD3 exclusively localizes in the cytoplasm to activate the MAP kinase signaling module and promote RAS-driven tumorigenesis via the methylation of MAP3K2 K260 [11,25]. Besides, SMYD3 was reported to methylate K14 of AKT1 and K175 of HER2 to enhance their respective activations through autophosphorylation in cancer cells [12,13]. Due to these important roles, targeting the methyltransferase activity of SMYD3 has been suggested to be a promising strategy for developing anti-cancer drugs.

We previously solved the crystal structure of the ternary complex of the SMYD3 protein, the SAH cofactor, and an MAP3K2 peptide (hereafter referred to as SMYD3-SAH-MAP3K2 complex) [26]. According to this analysis, SMYD3 has a two-lobed structure (Figure 1A). The N-terminal lobe contains a catalytic domain (SET and post-SET) including a lysine-binding channel and a SAM-binding pocket, and an MYND domain inserts into the SET domain. The C-terminal lobe is formed by a C-terminal domain (CTD) in proximity to the SET domain. By binding in the amphiphilic cleft on the surface formed by the N- and C-terminal lobes, the lysine residue of the substrate inserts into the lysine-binding channel to receive the methyl group from SAM in the SAM-binding pocket (Figure 1B). In the SAM-binding pocket, several residues including R14, N16, Y124, N132, N205, H206, Y257 and F259 hold the cofactor SAM or coproduct SAH using various interactions including hydrogen bonds as well as π-π and π-cation interactions (Figure 1C). In the lysine-binding channel, the three aromatic residues, namely F183, Y239 and Y257, form the narrow hydrophobic pocket to stabilize the sidechain of the lysine residue (Figure 1D). Substitutions of Y124, N132, F183, Y239 and F259 in the SAM-binding pocket and the lysine-binding channel decreased the catalytic activities of SMYD3 [27]. Besides, there is a shallow hydrophobic pocket on SMYD3, including S101, L104, V178, I179, S182 and V195, and which accommodates the binding of a phenylalanine residue at the −2 position of the substrate. Its significant role in serving as the determinant of the substrate specificity of SMYD3 was revealed by our previous experimental and computational studies [26,28].

The location of the unique CTD varies significantly between members of the SMYD family [2]. From a structural point of view, SMYD1 has an open conformation with a wide substrate-binding site, whereas SMYD3 forms a relative closed conformation with its clamshell-like structure (Appendix A) [29,30]. SMYD2 has an intermediate conformation between those of SMYD1 and SMYD3. Additionally, analysis of the crystal structures revealed that the CTD of SMYD2 undergoes a movement when different cofactors bind (Appendix A). It has been indicated that cofactors may have the ability to induce a conformational change of the CTD to affect substrate specificity in SMYD2 [31]. Although such allosteric properties are not directly found in the crystal structures of SMYD3, several biochemical and computational studies have provided evidence that the CTD may play a key role in regulating the catalytic activity. A previous work found that human cancer cells express a cleaved form of the SMYD3 protein that lacks the first 34 amino acid residues in the N-terminal region where the SAM-binding pocket is located. Interestingly, the truncated protein has been measured to have a higher catalytic activity than the full-length protein, suggesting that the above cleavage may induce a conformational change of the protein [32]. In another study, truncation of the whole CTD and a truncation of the last three C-terminal helices were each demonstrated to eliminate methylation of H4, indicating that the CTD is essential for SMYD3 methyltransferase activity [33].

Compared with experimental methods, computational research can overcome the time and space limitations to provide insights into the dynamical features at an atomic level. In a recent study, an open CTD conformation in SMYD3 was predicted in the absence of SAM according to molecular dynamics (MD) simulations. It was postulated that SAM acts like a key and that SMYD3 may undergo a closed-to-open conformational change without the binding of SAM [34]. Additionally, a similar investigation involving MD simulations and small-angle X-ray scattering revealed that SMYD3 can adopt an open conformation even in the presence of SAM [35]. Thus, the conformational dynamics of SMYD3 are still not completely clear. Further studies are necessary to explore the role of the SAM-binding pocket in modulating the CTD conformational change. Besides, the lysine-binding channel, which is located adjacent to the SAM-binding pocket, should also be considered by binary SMYD3-peptide or ternary SMYD3-cofactor-peptide complex.

To this end, in the current work, we carried out extensive MD simulations on a series of complexes with a total of 21 μs sampling to investigate the closed-to-open conformational changes of SMYD3 and unveil the molecular mechanisms behind the conformational dynamics in terms of structure and energy. A flowchart of the MD simulations was summarized in Appendix A. Starting from the crystal structures, six models were built (Appendix A), including unary SMYD3, binary SMYD3-SAM, binary SMYD3-MAP3K2, ternary SMYD3-SAM-MAP3K2, binary SMYD3-SAH, and binary SMYD3-GSK2807. GSK2807 is a reported inhibitor of SMYD3 and acts as a SAM analog, binding both in the SAM-binding pocket and the lysine-binding channel (Figure 1B) [36]. The chemical structures of SAH, SAM and GSK2807 are shown in Appendix A. For comparison purposes, these simulated models were denoted as Apo, SAM, MAP3K2, SAM_MAP3K2, SAH, and GSK2807, as shown in Table 1. Based on the CTD movements, simulated models in the current work were obtained in three different conformational states: the closed, intermediate and open states. Only in the case that both the SAM-binding pocket and lysine-binding channel were modeled to be bound did SMYD3 maintain its conformation in the closed state according to our simulations, indicative of a synergetic effect of the cofactors and the target lysine on regulating the conformational change of SMYD3. Several key residues were picked out to shed light on how the two regions regulate the CTD movement.

Besides, two additional simulated models, unary open_SMYD3 and ternary open_SMYD3-SAM-MAP3K2 (denoted as Apo_open and SAM_MAP3K2_open, Table 1 and Appendix A), were built and subjected to MD, as well as metadynamics simulations to further verify the conclusions. This mechanistic study provided insights into the relationship between the conformational change and the methyltransferase activity of SMYD3. The more complete understanding of the conformational dynamics that we here developed may along with further work lay a foundation for the rational drug design of SMYD3 inhibitors.

## 2. Materials and Methods

### 2.1. System Setup

In this study, a total of eight simulated models were separately subjected to MD simulations (Table 1). Among them, six models including unary SMYD3, binary SMYD3-SAM, binary SMYD3-MAP3K2, ternary SMYD3-SAM-MAP3K2, binary SMYD3-SAH, and binary SMYD3-GSK2807, used the same starting coordinates of SMYD3, specifically those derived from our previously solved crystal structure of the ternary SMYD3-SAH-MAP3K2 complex (PDB: 5EX0) [26] for the sake of comparison and evaluation. This crystal structure also supplied the starting structures of SAH and the MAP3K2 peptide in the simulations of systems including them. These components made up unary SMYD3 structure, binary SMYD3-MAP3K2 and SMDY3-SAM complexes in the Apo, MAP3K2 and SAH systems, respectively. For the binary SMYD3-SAM, ternary SMYD3-SAM-MAP3K2 and binary SMYD3-GSK2807 complexes in the SAM, SAM_MAP3K2 and GSK2807 systems, the starting coordinates of SAM and GSK2807 were obtained from the crystal structures of SMYD3 complexed with, respectively, SAM and GSK2807 (PDB: 3MEK, 5HI7) [36] by superimposing the protein. Besides, two additional systems, including Apo_open and SAM-MAP3K2_open, were setup using a modeled open-state structure. For the Apo_open model, the initial structure was built from an open conformational state, which was generated from the Apo system based on cluster analysis (see the following Section 2.3). For the SAM_MAP3K2_open model, the same initial structure as the Apo_open model was used, and SAM and MAP3K2 peptide were added back by superimposing with the known crystal structure 5EX0. The initial structure for each simulated model was presented in Appendix A.

After the construction of these structures, protonation states of histidine residues in the protein were assigned as predicted by H++ [37]. Three zinc ions in the crystal structure were retained and the chelating cysteine and histidine residues were deprotonated. Then, missing residue sidechains and hydrogen atoms were added to each complex by using the Leap module of the AMBER18 package. The AMBER ff14SB force field [38] was used to simulate the protein and MAP3K2 peptide, whereas the force field parameters used for SAM, SAH and GSK2807 were derived from the general AMBER force field (GAFF) [39]. The TIP3P water model [40] was used to solvate the complex in a hexagonal explicit water box under the periodic boundary condition. The distance between the edges of the box and any atom of the complex was at least 15 Å. NaCl was added to neutralize the system to obtain a salt concentration of 0.15 M. Details for each simulated system were presented in Table 1.

### 2.2. Molecular Dynamics Simulations

For each solvated system, a 5000-step energy minimization was performed to remove conflicts and overlaps between atoms (Appendix A). Then, the complex was equilibrated for 500 ps in a constant-volume ensemble to heat the system from 0 to 300 K, followed for 500 ps in a constant-pressure ensemble, specifically at 1 bar. The Langevin thermostat [41] and the Berendsen barostat [42] were used for temperature and pressure controls, respectively. During equilibration, a force constant of 20 kcal mol^−1^ Å^−2^ as a harmonic constraint was applied to the complex with 1-fs timestep. The temperature and density of each simulated system were approximate 300 K and 1 g/cm^3^ after the equilibration stage, respectively (Appendix A).

With the constraint released, MD simulation was performed in constant-pressure ensembles at 300 K. For the first six and last two systems in Table 1, three independent parallel MD simulations with different initial velocities were carried out each for 1 and 0.5 μs, respectively. The potential was calculated using the Hamiltonian function [43] (see Materials and Methods section in Appendix A). Periodic boundary conditions were applied in all three directions. The cut-off value of the van der Waals interactions was set to 12 Å. The particle mesh Ewald (PME) method [44] was used to calculate the long-range electrostatic contributions. The SHAKE algorithm [45] was used to restrain all of the bond lengths involving hydrogen atoms. The timestep was set to 2 fs for each system during the MD production stage. The AMBER18 software package [46] was employed to carry out MD simulations. Coordinates were saved every 10 ps.

### 2.3. Analysis of the Simulations

The cpptraj module of AmberTools18 was used to calculate the conformational descriptors, key hydrogen-bonding lifetime and dynamic cross-correlation map (DCCM) along each MD simulation. The conformational descriptors included root mean square deviations (RMSDs), root mean square fluctuations (RMSFs), radius of gyration (RoG), defined distances D1, D2 and D3, and the dihedral angle of Y239. RMSD was calculated by the equation,
(1)$$\mathrm{RMSD}=\sqrt{\frac{\sum_{i=0}^{N}\left[m_{i}*{\left(X_{i}-Y_{i}\right)}^{2}\right]}{M}}$$
where *N* is the number of atoms, $m_{i}$ is the mass of atom *i*, $X_{i}$ and $\;Y_{i}$ are the coordinate for target atom *i* and reference atom *i*, respectively, and *M* is the total mass. The distance D1 was measured between the centroid of residues 42–48 of the MYND domain and the centroid of residues 298–302 of the CTD. The distance D2 was measured between the centroid of residues 209–227 of the SET domain and the centroid of residues 363–365 of the CTD. The distance D3 was measured between the sidechain-centroids of residues 183 and 239. The dihedral angle of Y239 measured was specifically the CA-CB-CG-CD2 torsion angle of this tyrosine. For each simulated system, the protein or the complex from the first trajectory of three parallel MD simulations was subjected to the structural analyses. The illustrated representative conformation was derived from the center frame of cluster analysis based on its RoG calculation. The representative open state in the Apo system was used as the initial structures to build the Apo_open and SAM_MAP3K2_open systems.

### 2.4. Potential of Mean Force

The potential of mean force (PMF) of each system was calculated along with RMSD and distance D1 throughout the MD simulation, and depicted the conformational changes due to the adequate sampling applied. The energy landscape was calculated using the equation
(2)$$\Delta G\left(x,y\right)=k_{B}T\ln g\left(xy\right)$$
where *k_B_* is the Boltzmann constant, *T* is the simulation temperature, and *g*(*x*, *y*) is the normalized joint probability distribution. The minimum energy was set to zero. A bin size of 0.1 Å was used for the RMSDs and distance D1. A total of 100,000 and 50,000 frames were used for the PMF calculations for the first six and last two systems, respectively.

### 2.5. Binding Free Energy

The binding free energies and residue decompositions between the SET domain and the CTD were calculated using the MM/GBSA method [47,48,49]. A total of 1000 snapshots were extracted from the final 100 ns trajectory of each system for calculation. The SET and MYND domains (residues 1–241) were considered as the receptor, and the CTD (residues 270–428) was considered as the ligand. All of the parameters were set as default values in the calculations. Since we were mainly interested in the differences of the residue-decomposed binding free energies, entropy was ignored in the calculations.

### 2.6. Metadynamics Simulation

Four simulated systems, Apo, Apo_open, SAM_MAP3K2 and SAM_MAP3K2_open, were subjected to the well-tempered metadynamics simulations [50,51,52], which were performed using the AMBER18 package [46] and PLUMED plugin [53]. For each system, three independent parallel metadynamics simulations with different initial velocities were carried out after equilibration stage for 50 ns. The distance D1 was chosen as the collective variable (CV). Based on time evolutions of the D1 distances of the simulated systems, the upper and lower limits of D1 were set to 20 and 6 Å, respectively. The force constant was set to 5000 kJ·mol^−1^·Å^−1^. The upper and lower bounds for the grid were set to 25 and 5 Å, respectively. The metadynamics simulation was activated in CV by depositing a Gaussian bias term every 1 ps with height of 1.0 kJ·mol^−1^ and width of 0.5 Å. The bias factor was 15, and the temperature was 300 K. A total of 50,000 frames were used for calculating the free energy landscape for each system. The temperature in energy unit for integrating out variables was set to 2.5 kJ·mol^−1^.

## 3. Results and Discussions

### 3.1. Overall Conformational Changes during MD Simulations

Conformational changes of the CTD in each of the six systems were monitored by carrying out a series of measurements (Figure 2). As shown in Figure 2A and Appendix A, the radius of gyration (RoG) of the protein during three 1 µs parallel simulations in each system were calculated, and were used to estimate the potential for each protein to buckle. Results of the three parallel simulations in each system were quite identical to each other, and the results from the first parallel simulation were subjected to analyses and discussions unless otherwise specified. In these simulated models, the average values of RoG were measured to be in the range 22.66 to 22.95 Å (Appendix A), indicating different conformational states. Additionally, the Apo, SAM and SAH systems showed much higher standard deviations than did the MAP3K2, SAM_MAP3K2 and GSK2807 systems, implying large conformational changes or conformational transitions in the former systems. Small angle X-ray Scattering (SAXS) experiments were further carried out to validate the MD simulation results. Four groups of protein or protein complexes, including Apo, SAM, MAP3K2 and SAM_MAP3K2 were prepared for SAXS measurements. Though SMYD3 showed slightly aggregations by X-ray radiation as the upturns at low-q region of Guinier plots were observed and the RoG values increased with the increasing of protein concentrations within the same group of samples (Appendix A and Appendix A), the SAXS results is probably reliable according to previous SAXS reports on SMYD3 [35]. In any case, SAXS samples with same protein concentration exhibit larger RoG values in the Apo and SAM groups than those in the MAP3K2 and MAP3K2_SAM groups, which were coincide with the MD simulations results (Figure 2A and Appendix A).

As the RoG analyses showed the biggest difference between the apo-SMYD3 structure in the Apo system and the ternary SMYD3-SAM-MAP3K2 complex in the SAM_MAP3K2 system, we paid attention to the mobile regions of the two systems by applying per residue root mean square fluctuation (RMSF) analyses. As shown in Figure 2B, similar RMSF values were found in both systems, except for two distinct differences in the CTD. In the Apo system, residues 279–318 in the first two anti-parallel α-helices of the CTD (highlighted in the red box in Figure 2B) showed higher fluctuations in the calculations than did residues in the SAM_MAP3K2 system, whereas residues 358–372 at the turn between the fourth and fifth α-helices of the CTD (highlighted in the black box in Figure 2B) showed lower flexibility.

Due to the RMSF analyses indicating different motions of the CTDs of the two systems, a cross-correlation analysis was performed to analyze the motions of those residues and the conformational changes of the CTD. Figure 2C shows the calculated dynamic cross-correlation maps of the Apo (upper left triangle) and SAM_MAP3K2 (lower right triangle) systems. The correlation coefficients of the different regions of the protein are shown in different colors. The correlated motions involving the residues labeled in Figure 2B are highlighted by the same scheme. In the Apo system, negative correlation coefficients were calculated for residues 358–372 of the CTD and residues 43–70 of the MYND domain (highlighted in the red box in Figure 2C), indicative of a movement of the CTD away from the MYND domain; whereas in the SAM_MAP3K2 system, positive correlation coefficients of these residues were calculated, indicative of strong correlated motions of these regions. By contrast, generally higher residue–residue correlations between the CTD (residues 358–372) and the SET domain (residues 206–233) were found for the Apo system than for the SAM_MAP3K2 system (highlighted in the black box in Figure 2C), indicating a weaker coupling of these regions in the SAM_MAP3K2 system than in the Apo system.

To better demonstrate and characterize the CTD dynamics, two distances were calculated, one located at the top of the substrate-binding cleft between the MYND domain and the CTD and denoted as D1, and the other at the bottom of the substrate-binding cleft between the SET domain and the CTD and denoted as D2 (Figure 2D–F, Appendix A, Appendix A, and Experimental Procedures). Time evolutions of the D1 distances of the simulated systems (Figure 2E) showed trends very similar to those observed for the RoG (Figure 2A). The highest displacements of the CTD away from the MYND domain, with an average value as high as 12.07 Å and a maximum value as high as 20.73 Å, were found for the apo-SMYD3 structure in the Apo system. Analysis of the structure of SMYD3 complexed with SAH in the SAH system showed it to have the second-greatest CTD-MYND distance, with an average value of 10.71 Å. Although having a broader distribution with a higher standard deviation, the structure of SMYD3 complexed with SAM in the SAM system exhibited a little lower CTD displacement, with an average value of 9.88 Å, than the binary SMYD3-MAP3K2 complex in the MAP3K2 system with an average value of 10.11 Å. As expected, the ternary SMYD3-SAM-MAP3K2 complex in the SAM_MAP3K2 system showed low fluctuations with a low average CTD displacement value of 8.43 Å, much closer to the initial value of 8.93 Å in the crystal structure compared to the other systems. While in the case of the structure of SMYD3 complexed with GSK2807 in the GSK2807 system, the time-evolved variations were observed to be very similar to those of the SAM_MAP3K2 system with an average displacement value of 8.38 Å. Thus, the distance D1 measurements gave a ranking of Apo > SAH > MAP3K2 > SAM > SAM_MAP3K2 ≈ GSK2807.

Figure 2F shows the time evolution of distance D2. An initial value of 8.11 Å was calculated for the crystal structure. Average values of 7.29, 8.02, 7.74, 9.39, 9.09, and 9.85 Å were calculated for D2 in the simulated systems, respectively. That is, the D2 measurements gave a ranking of GSK2807 > SAM_MAP3K2 > SAH > SAM > MAP3K2 > Apo, more or less the reverse of that shown for the distance D1. Additionally, the time evolutions of the distance D2 in the Apo, SAM and SAH systems showed a trend of fluctuations obviously opposite that for the distance D1 in those systems (Figure 2E,F). These distance measurements involving, as described above, the top and bottom of the substrate-binding cleft for D1 and D2, respectively, thus suggested that the CTD undergoes clamp-like conformational changes, with the amount of space at the top inversely associated with the amount of the space at the bottom.

### 3.2. Conformational States in the Simulated Systems

To gain more insight into the change in energetics accompanying the conformational change of the CTD, the potential of mean force (PMF) in each system was calculated to depict an energy landscape based on the large sampling space. In order to obtain a two-dimensional energy landscape map, the distance D1 and the root mean square deviations (RMSDs) of the protein were chosen as the reaction coordinates to monitor the structural change. The time evolutions of RMSD and the average values for the simulated systems are shown in Appendix A and Appendix A.

Inspection of the PMF maps clearly suggested the ability of SMYD3 to access three different conformational states, and to adopt them at different relative frequencies for its different cofactor- and/or substrate-binding statuses. The average D1 distance values of the three states were measured to be approximately 8 to 9, 10 to 11, and 13 to 14 Å, indicating the closed, intermediate and open states, respectively (Figure 3A). The starting crystal structure of the protein was observed to be in the closed state with an initial distance value of 8.93 Å. As for the simulated models, 53.63% and 46.37% of the population of the protein in the Apo system occupied the intermediate and open states, respectively (Appendix A). The D1 measurements (Figure 2E) exhibited an extremely unstable structure with frequent conformational transitions between the two states. With the cofactor SAM bound, i.e., in the SAM system, the occurrences of the intermediate and open states decreased to 40.29% and 12.93%, respectively, whereas 46.78% of the population of the protein occupied the closed state. In contrast, the binding of the MAP3K2 peptide instead of SAM, i.e., to form the MAP3K2 system, resulted according to the simulations in only the intermediate state. The binding of both SAM and MAP3K2 peptide to SMYD3 to form the SAM_MAP3K2 system was indicated to result in only the closed state (as observed for the referenced crystal structure), and hence a stable ternary complex. Similar to the binary SMYD3-SAM complex, SMYD3 complexed with SAH in the SAH system accessed all three conformational states in the simulations—specifically with occupancies of 49.35%, 16.75% and 33.90% for the closed, intermediate and open states, respectively, implying a more flexible CTD movement because of the higher occupancy remaining in the open state compared to that for the SAM system. These results thus indicated some differences between the roles of cofactors SAM and SAH in regulating the CTD conformational dynamics. Finally, like SMYD3 in the SAM_MAP3K2 system, SMYD3 when complexed with GSK2807, i.e., in the GSK2807 system, was indicated to only occupy the closed state. GSK2807 was previously shown to bind both in the SAM-binding pocket and the lysine-binding channel of SMYD3 [36], but to not interact directly with the CTD, in contrast to the observation of an interaction of the MAP3K2 peptide of the ternary complex in the SAM_MAP3K2 system with the CTD. Thus, the effect of direct interactions between the substrate and the CTD on the closed conformational state could be excluded. In addition, for each system, the conformational states depicted by PMF maps in three parallel simulations were similar (Figure 3A and Appendix A), indicating that the samplings properly converged after long MD simulations, especially for the flexible models such as Apo, SAM and SAH. Taken together, from the analyses of the PMF maps, the Apo system was concluded to adopt a mixture of open and intermediate conformational states, the SAM and SAH systems each a mixture of open, intermediate and closed states, the MAP3K2 system the intermediate state, and the SAM_MAP3K2 and GSK2807 systems each the closed state.

To illustrate the open, intermediate and closed states, the representative conformations were chosen from the Apo, MAP3K2 and SAM_MAP3K2 systems, respectively (Figure 3B). A pronounced outward movement of the CTD was observed from the closed to open state, leading to an expansion of the substrate-binding crevice. From combining analyses of the PMF maps and representative conformations, the closed and intermediate states were concluded to be steady states, in which the protein could stably form a direct lobe–lobe interaction between the MYND domain and the CTD in the MAP3K2, SAM_MAP3K2 and GSK2807 systems. By contrast, the open state was concluded to be an unsteady state due to the protein here lacking the equivalent interaction. Thus, the protein was indicated, based on the distance measurements in the Apo, SAM and SAH systems, to undergo frequent conformational transitions between the intermediate and open states.

### 3.3. The Key Residues Involved in the CTD Movement

We set out to unveil the molecular mechanisms behind the transformation from the closed to open conformations of SMYD3 by analyzing their structural details. For this purpose, hydrogen bond analyses were performed to determine the key residues for both conditions. We listed the hydrogen bonds having different occupancies in the different simulated systems. In this way, five pairs of hydrogen bonds were selected, namely R14-D262, M242-R265, A188-H404, S44-V193 and K42-E295. Table 2 and Appendix A shows the occupancies of these hydrogen bonds in each parallel simulated model with an occupancy of a hydrogen bond defined as the percentage of the time during the simulation that the bond is formed. To elucidate their roles in inducing the CTD movement, the representative conformations from the Apo and SAM_MAP3K2 systems, i.e., those corresponding to the closed and open states, were used to illustrate the hydrogen bond states at their respective positions (P1–P5 in Figure 4A).

In the apo-SMYD3 structure, the frequent CTD movements would be expected to induce the formation of a noncompact structure and an unstable interface between the SET domain and the CTD. One of the reasons for this flexibility may be derived from the SAM-binding pocket. Without the binding of SAM, residue R14 would become too flexible to form a hydrogen bond with residue D262, and indeed, our simulations showed here an occupancy of only 1.76% for this hydrogen bond (Figure 4B, Table 2). The unstable SAM-binding pocket was indicated by our simulations to influence the inner p-SET domain M242-R265 hydrogen bond, which also showed a low occupancy, of 13.70% (Figure 4C, Table 2). An exception to this pattern in the apo-SMYD3 structure was found to be the A188-H404 hydrogen bond at the interface of the SET domain and the CTD, with this bond showing a high occupancy of 63.17%, attributed to the CTD movement during the MD simulation (Figure 4D, Table 2). Finally, the S44-V193 hydrogen bond at the interface of the SET and MYND domains, and the K42-E295 hydrogen bond at the interface of the MYND domain and the CTD exhibited low occupancies of 21.49% and 5.73%, respectively, indicative of a break in the direct lobe–lobe interaction (Figure 4E,F and Table 2).

In the ternary complex, which remained in the closed conformational state during the MD simulation, the occupancies of the hydrogen bonds were found to be quite different from those of the apo-SMYD3 structure. In the SAM-binding pocket, the adenosine group of SAM would appear to stabilize residue R14 by a π-cation interaction and to lead to a stable R14-D262 hydrogen bond, and indeed an occupancy of 90.83% was calculated for this bond (Figure 4B, Table 2). This simulation result indicated a large role played by SAM in stabilizing the SAM-binding pocket. Additionally, the M242-R265, S44-V193, and K42-E295 hydrogen bonds were also indicated to be much more stable with higher occupancies of, respectively, 48.38%, 87.23% and 50.51% in the SAM_MAP3K2 system than in the Apo system (Figure 4C,E,F and Table 2). In contrast, the A188-H404 hydrogen bond, having a long lifetime in the simulation of the Apo system, was hardly detected in the simulation of the SAM_MAP3K2 system, showing here an occupancy of only 3.43% (Figure 4D, Table 2).

In addition, the results from other simulated systems and parallel simulations shown in Appendix A also supported the assumption above. Correlations between the binding of SAM and the lifetimes of the R14-D262 and M242-R265 hydrogen bonds were observed. Their occupancies were calculated to be much lower in the Apo and MAP3K2 systems than in other systems. The weakened natures of these hydrogen bonds in these SAM-free systems were attributed to the locations of these hydrogen bonds, respectively, in and near the SAM-binding pocket. Moreover, due to the proximity of the M242-R265 hydrogen bond to the interface of the SET domain and the CTD, fluctuations induced by the breaking of the hydrogen bond may disturb this interface. Correlations between the conformational states of SMYD3 and the lifetimes of the A188-H404, S44-V193 and K42-E295 hydrogen bonds were also observed. The three hydrogen bonds are located at the interface of their respective domains (i.e., the SET domain and the CTD, the SET and MYND domains, and the MYND domain and the CTD). In the simulations, as the CTD moved away from the SET domain during the transformation from the closed to open state, the occupancies of the V193-S44 and K42-E295 hydrogen bonds decreased, but the occupancy of the A188-H404 hydrogen bond increased, indicative of a significant role of these hydrogen bonds in determining the different conformational states. In addition, note the consistency between the changing trends of the three occupancies and the monitored distances D1 and D2, and this consistency providing further support for the occurrence of clamp-like conformational changes. In conclusion, the CTD movement may be induced by the binding of SAM through the monitored hydrogen-bond network.

### 3.4. Investigations of the Lysine-Binding Channel

The size of the lysine-binding channel of the protein was characterized using the defined geometric descriptor distance D3 (Figure 5A and Experimental Procedures). The initial value of D3 was calculated from the crystal structure to be 7.70 Å. Average D3 values of 6.20, 7.03, 7.07, 7.50, 7.21 and 8.56 Å were calculated in the simulated systems, respectively, giving a rank of Apo < SAM ≈ MAP3K2 < SAH < SAM_MAP3K2 < GSK2807 (Figure 5B and Appendix A, Appendix A). These calculations indicated the occurrence of some changes in the channel upon the binding to this channel of cofactor or substrate. In the Apo system, a representative apo-SMYD3 structure of the simulation showed Y239 able to adopt a conformation different than that in the crystal structure. In this simulation of the apo state, i.e., without the target lysine bound, residue Y239 adopted a side chain conformation bringing it closer to residue F183 (Figure 5C). While in the SAM_MAP3K2 system, i.e., in the ternary SMYD3-SAM-MAP3K2 complex, residue Y239 of SMYD3 showed relatively little conformational freedom due to its interactions with residue K260 of MAP3K2, and hence maintained the same sidechain position as initially in the crystal structure (Figure 5D).

To further investigate the sidechain conformation of residue Y239, its dihedral angle was measured by specifically measuring its CA-CB-CG-CD2 torsion angle (hereafter referred to as dihedral Y239). As shown in Figure 5E and Appendix A, in the MAP3K2, SAM_MAP3K2 and GSK2807 systems, the average values of dihedral Y239 were measured to be very close to the initial value in the crystal structure (183.11°), with narrow spreads. While in the other systems, especially the Apo and SAH systems, the dihedral angles were calculated to have wide distributions. These results were consistent with a poor ability of the sidechain of Y239 to rotate freely with lysine bound in the lysine-binding channel. In addition, a positive correlation between the distributions of the dihedral angle and the fluctuations of the CTD movements was found by comparing the standard deviations of the dihedral Y239 and the distance D1 (Appendix A). The flexibility of CTD of SMYD3 may be decreased by the target lysine on the substrate being inserted into the lysine-binding channel.

Combining the distance and dihedral angle measurements led to additional deep insights. The lysine-binding channel comprises three aromatic residues that form a core around the target lysine (Figure 1D). Due to energetic costs of exposing a hydrophobic environment to solvent, residue Y239 without a target lysine bond would be expected to undergo a conformational change to decrease its solvent accessible surface area, resulting in a shrunken channel. Besides, compared with the D3 distance values in the other simulated models, D3 in the binary SMYD3-GSK2807 complex was measured to be larger, with a narrower distribution (Figure 5B). GSK2807 was designed as a SAM analog, to introduce a propyl dimethylamine sidechain to mimic a dimethylated lysine (Appendix A). Indeed, the size of the lysine-binding channel was observed in the simulations of the binary SMYD3-GSK2807 complex to be enlarged by enough in order to accommodate the two extra methyl groups on the sidechain of GSK2807. These results demonstrated the self-adjustment ability of the lysine-binding channel to accommodate target lysines with different methylation states.

In addition, the distributions of dihedral Y239 in the binary SMYD3-SAM and SMYD3-SAH complexes showed obvious differences (Figure 5E). In the SAM system, time evolutions of the dihedral angle showed the sidechain of residue Y239 stabilizing at either of two conformations, the initial conformation (~180°) or the flipped conformation (~270°). However, in the SAH system, the sidechain of residue Y239 frequently flipped back and forth between the two conformational states (Appendix A). These results indicated cofactor SAM to not only function to bind into the SAM-binding pocket, but to also have a role in regulating the conformation of residue Y239 in the lysine-binding channel. Compared with SAH, SAM has an extra methyl group that orients to the lysine-binding channel to interact with the target lysine. As shown in Figure 5F, the methyl group on SAM was indicated from our analyses to sterically interfere with the conformational transition of residue Y239, although while hardly maintaining residue Y239 in the initial conformational state as in GSK2807, in which the sidechain was observed to be bound into the lysine-binding channel.

### 3.5. Lobe–Lobe Interface Energy Analyses

The SET and MYND domains in the N-lobe have been found to directly interact with the CTD of the C-lobe in the closed conformational state of SMYD3, but to have this interaction broken in the conformational transition from the closed to open state. Therefore, in each simulated system, the binding free energy and residue decompositions at the lobe-lobe interface during the last 100 ns of the MD simulation were calculated in order to describe the conformational dynamics of SMYD3 in terms of energy (Table 3). As expected, the N-lobe in the ternary SMYD3-SAM-MAP3K2 complex was calculated to have the most negative binding free energy of −32.41 kcal/mol toward the C-lobe, whereas for the binary SMYD3-GSK2807 complex, this binding free energy was about −30.91 kcal/mol. Both complexes were sampled only in the closed conformation during the MD simulations. These results indicated the direct lobe–lobe interaction in the closed conformation to be strong enough to make a great contribution to the compact and stable structure. The binary SMYD3-MAP3K2 complex, which formed an intermediate state during the whole MD simulation, was calculated to have an intermediate binding free energy of −18.90 kcal/mol, indicative of a weak lobe–lobe interaction. As to the other three simulated models, based on the D1 distance measurements (Figure 2E), we speculated the apo-SMYD3 structure and the binary SMYD3-SAM complex to be in a hybrid state of intermediate and open states in their respective last 100 ns of the MD simulation, but the binary SMYD3-SAH complex to be in the intermediate state. The binding free energies in these three systems were calculated to be −17.72, −15.77 and −16.93 kcal/mol, respectively. The high similarity of the binding free energies of the Apo, SAM, MAP3K2 and SAH systems indicated a low potential barrier between the intermediate and open state, thus explaining why the open state was sampled along with the intermediate state during the simulations.

By calculating residue-decomposed binding free energies, several residues were picked out as those for which the difference of the decomposed binding free energy at the corresponding position in any of the simulated systems is more than 1 kcal/mol (Appendix A). Figure 6 shows these residues in the representative conformations of the SAM_MAP3K2 and Apo systems. For comparison purposes, they are colored by their respective decomposed binding free energies. In the closed state (Figure 6A), residues K42, G43 and S44 in the MYND domain, and residues E295, H299 and W300 in the CTD domain were observed to be involved in the direct lobe–lobe interaction—with the interaction forming a hydrophobic environment, suitable for accommodating the aromatic sidechain of residue W300. In this situation, the K42-E295 hydrogen bond would be expected to be stable due to the low dielectric constant of the hydrophobic environment. In addition, the stable and compact structure was observed to enclose residue H404 in the hydrophobic environment. Thus, these residues would be expected to make great contributions to the lobe–lobe interaction and structural stability. An exception might be residue P363. The D2 distance measurements showed an increased distance in the closed state (Figure 2F) at the location of residue P363, and hence resulting in residue P363 making few contributions to the binding in this state.

In a transition to the open state (Figure 6B), the CTD outward movement would break the direct lobe–lobe interaction, leading to hydrophobic residues H299 and W300 becoming exposed to the solvent. In addition, the CTD movement would also destabilize the hydrogen bond K42-E295, resulting in unneutralized charges at the lobe–lobe interface. Thus, residues K42, G43, S44, E295, H299 and W300 would here make few contributions to the binding. However, residue H404 was calculated to have a much lower decomposed binding free energy in the open state than in the closed state, attributed to the long-lifetime A188-H404 hydrogen bond in the open state (Table 2). The distance D2 measurements showed little variation in the open state (Figure 2F). The hydrophobic environment was reasoned to lead to the calculated negative decomposed binding free energy of residue P363.

### 3.6. Free Energy Landscapes of the Conformational Changes

To obtain a further understanding of the conformational change of SMYD3 induced by cofactor and substrate binding, we set out to examine whether the open state could reverse back to the closed state when both the SAM-binding pocket and lysine-binding channel were bound. Thus, two additional systems, Apo_open and SAM_MAP3K2_open, were built based on the same open conformational state sampled from the Apo system and subjected to MD simulations (Table 1 and Experimental Procedures). Based on the measurement of the distance D1 and the RMSD of SMYD3 (Appendix A), the conformational states during three parallel simulations in each system were similar as depicted by PMF maps (Figure 7A,B and Appendix A). In each of the parallel Apo_open systems, the distance D1 had a large variation range of 10 Å to 20 Å caused by the flexible CTD movement. The PMF maps showed frequent conformational transitions between the intermediate and open states, which were consistent with those in the Apo system. However, in the SAM_MAP3K2_open system, the conformational transitions were different from the Apo_open system. The distance D1 decreased dramatically from an initial value of 13.82 Å and stabled at a range of ~9–10 Å after ~200 ns in each parallel MD simulation (Appendix A), which is very close to the value of 8.93 Å of the closed state in the crystal structure. These results suggested that the open state could be reversed back to closed state, or at least to the intermediate state when both the sites of cofactor and substrate were filled, and the closed state accounted for a larger portion of the conformational population in the doubly bound constraint (Figure 7B and Appendix A).

Except for the binding free energy of the lobe–lobe interface, metadynamics simulations were performed to depict free energy surface associated with the conformational change of SMYD3 from the perspective of the whole protein system (Figure 7C,D). Three parallel simulations for each system were carried out and the three landscapes in each system were very similar to each other (Appendix A). For the systems with the same components (Apo and Apo_open, or SAM_MAP3K2 and SAM_MAP3K2_open), the free energy surface landscapes were similar to each other no matter the initial structure adopted the open or closed conformation. Whereas for systems with different components, the curves demonstrated distinct properties. In the Apo and Apo_open system, the basin area was wide and flat and the free energy surface was stable from approximate 10 Å to 20 Å of the distance D1 (Figure 7C). This range of D1 corresponded to the intermediate and open states in the PMF maps (Figure 3A and Figure 7A). As there was no obvious potential barrier in the landscape, the two conformational states (open and intermediate) transited to each other frequently. However, in the SAM_MAP3K2 and SAM_MAP3K2_open system, the landscape showed one minimum value of free energy at ~9 Å of D1, which corresponded to the closed state and could not be observed in the unary SMYD3 systems (Figure 7D). Additionally, the free energy surface increased with the increasing of the distance D1. These results may explain why the ternary SMYD3-SAM-MAP3K2 complex could finally stabilize at the closed state where the unary SMYD3 could not, and put more evidence on the assumption that SMYD3 tends to adopt a closed conformation with constraints at both the cofactor and substrate binding pockets.

## 4. Conclusions

SMYD3 was previously confirmed to play a central role in a variety of cancer diseases [14,15]. Targeting SMYD3 and its function in the initiation and progression of cancer would support the discovery of anti-cancer drugs. However, many aspects of SMYD3 have yet to be understood, including its conformational changes and molecular mechanism. In this work, eight systems each with SMYD3 and up to two additional components were constructed and subjected to three parallel MD simulations to comprehensively investigate the conformational dynamics of the CTD of SMYD3. The simulated models sampled three different conformational states: the closed, intermediate and open states. When neither of the two additional components was bound (the Apo and Apo_open systems), the MD simulations showed SMYD3 undergoing frequent conformational transitions between the intermediate and open states. When only the SAM-binding pocket was filled with a bound component (the SAM and SAH systems), the simulations showed SMYD3 transitioning between the conformations of all three states. When only the lysine-binding channel was filled with a bound component (the MAP3K2 system), SMYD3 was observed to remain in the intermediate state. By contrast, when both of them had bound components (the SAM_MAP3K2, SAM_MAP3K2_open and GSK2807 systems), the simulations showed SMYD3 stabilizing at its closed-state conformation. In addition, different conformational states between the unbound and doubly bound systems were depicted by the free energy surfaces through metadynamics simulations. These results indicated that the cofactors in the SAM-binding pocket and the target lysine in the lysine-binding channel would have synergetic effects on the regulation of the CTD movement of SMYD3.

Molecular mechanism insights were provided by exploring the key residues involved in the conformational change. The lysine-binding channel was determined to have the ability to adjust itself, in particular its size, in order to accommodate the target lysine. The apo-SMYD3 structure without the target lysine inserted showed a relatively small channel, with the sidechain of residue Y239 undergoing frequent conformational transitions between the initial and flipped states. Additionally, without SAM bound in the SAM-binding pocket, the sidechain of residue R14 was indicated to be too flexible to form a stable hydrogen bond with residue D262, resulting in D262 fluctuating frequently. These two unstable regions were posited to regulate the formations of a series of hydrogen bonds at the interface between the N- and C-lobe, finally leading to the CTD movements and the clamp-like conformational change.

Analyses of the three observed conformational states supported the suggestion that the hinge motion of the CTD may regulate substrate recruitment and binding. SMYD3 has been shown to display a bi-bi random mechanism, i.e., a random order of the binding of the cofactor and the substrate to SMYD3 [54]. Thus, in the absence of bound substrate, and regardless of whether or not SAM is bound, SMYD3 was indicated in the current work to undergo a conformational change to the open state to widen the pocket for the substrate binding. Analyses of the geometric descriptors indicated this state to not be a steady state. Initial binding of the substrate, however, coincided in the simulations with a change in the SMYD3 conformation into the intermediate state. The binding free energy calculations indicated this state to be a structurally steady state but not a minimum-energy state. Only with both SAM and substrate bound did SMYD3 in the simulations maintain its conformation in the closed state, which is the structurally and energetically steady state. The free energy difference between the closed and intermediate state was calculated to be about 14 kcal/mol. The low-energy conformation could facilitate the methylation process and improve catalysis efficiency. In summary, this mechanistic study provided insights into the relationship between the conformational change and the methyltransferase activity of SMYD3. A more complete understanding of the conformational dynamics may lay a foundation for the rational drug design of SMYD3 inhibitors.

## Figures and Tables

**Figure 1 ijms-22-07185-f001:**
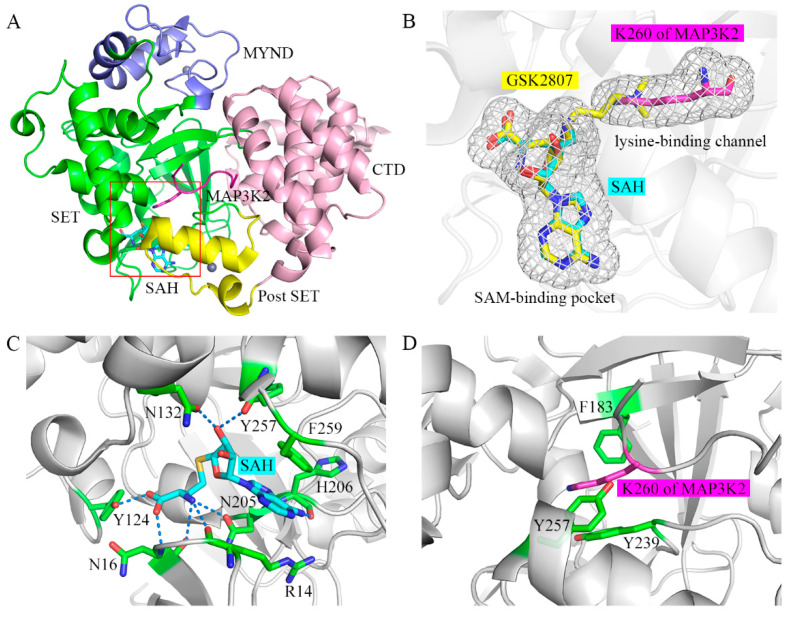
Overall structure of the SAM-binding pocket and the lysine-binding channel in SMYD3. (**A**) Crystal structure of SMYD3 in complex with substrate peptide and cofactor SAH (PDB: 5EX0). The SET, MYND, post-SET, and C-terminal domains of SMYD3 are shown in green, blue, yellow, and pink, respectively. SAH and the MAP3K2 peptide are shown as cyan and magenta sticks, respectively. Zinc ions are shown as grey spheres. The SAM-binding pocket and the lysine-binding channel are highlighted in the red box. (**B**) Magnified view of the SAM-binding pocket and the lysine-binding channel. GSK2807 is shown as yellow sticks. (**C**) Binding mode of SAH in the SAM-binding pocket. Residues in SMYD3 are shown as green sticks. The hydrogen bond is depicted as a dashed line. (**D**) Binding mode of the target lysine residue in the lysine-binding channel.

**Figure 2 ijms-22-07185-f002:**
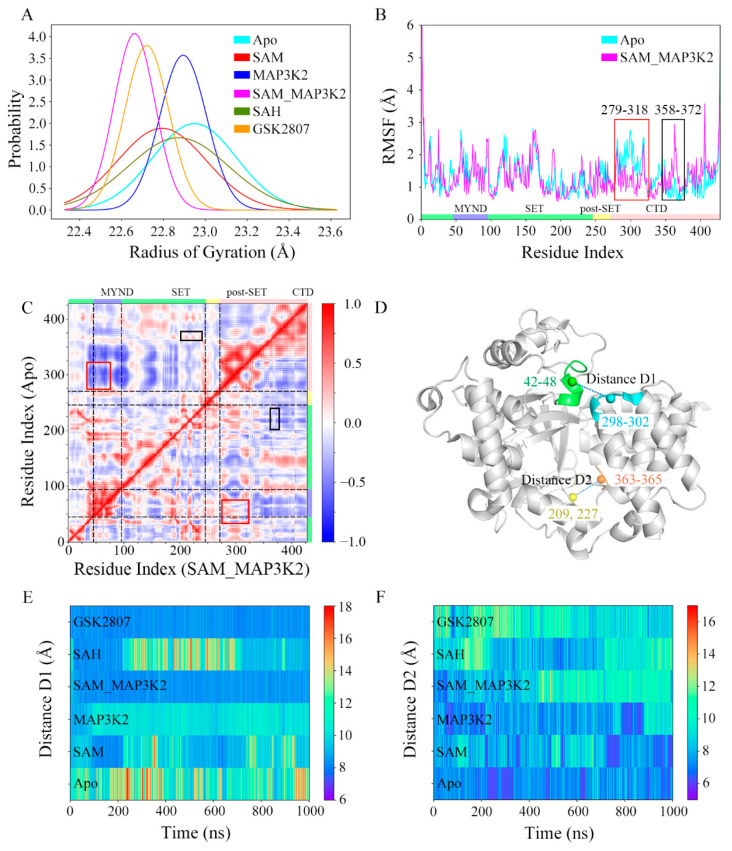
Conformational descriptors characterizing the dynamics of the CTD in SMYD3. (**A**) Distributions of RoG. The simulated systems are shown in cyan, red, blue, magenta, green and orange. The same scheme is used in the following figures unless otherwise specified. (**B**) RMSF analyses of the Apo and SAM_MAP3K2 systems. Two distinct differences are highlighted in the red and black boxes. (**C**) Dynamic cross-correlation maps for the Apo (upper left triangle) and SAM_MAP3K2 (lower right triangle) systems. The color scale is shown on the right changing from red (highly positive correlations) to blue (highly negative correlations). (**D**) Definition of distance D1 and D2 in the structure of SMYD3. Residues 42–48, 298–302, 209 and 227, and 363–365 are shown in green, cyan, yellow and orange, respectively. (**E**) Time evolutions of distance D1. The color scale is shown on the right changing from red (long distance) to blue (short distance). (**F**) Time evolutions of distance D2.

**Figure 3 ijms-22-07185-f003:**
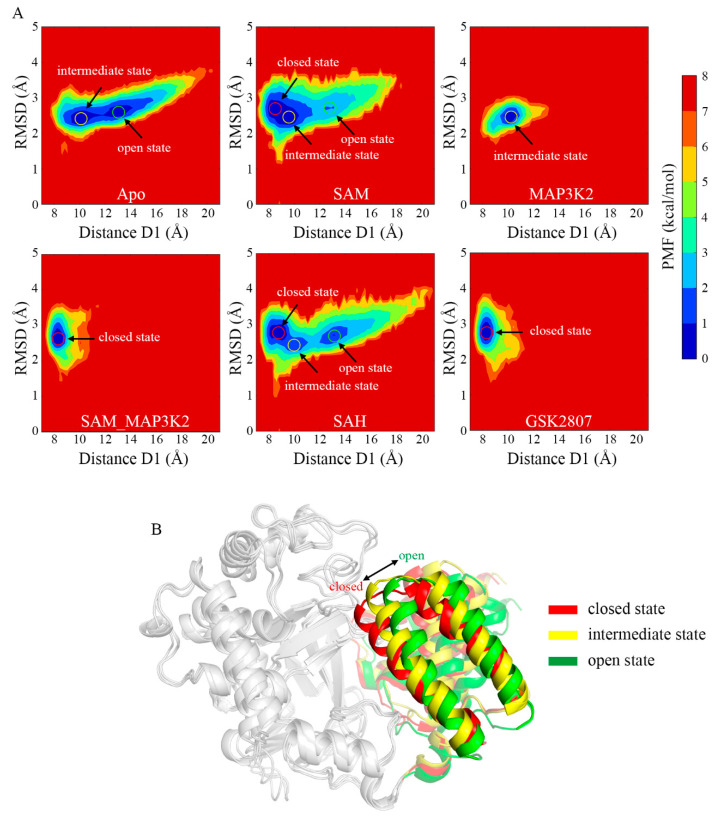
Conformational states of SMYD3 during MD simulations. (**A**) PMF calculated for the distance D1 vs. the RMSD of SMYD3. (**B**) Representative structures of the closed (red), intermediate (yellow), and open (green) states. Only the CTD is colored for comparison purposes.

**Figure 4 ijms-22-07185-f004:**
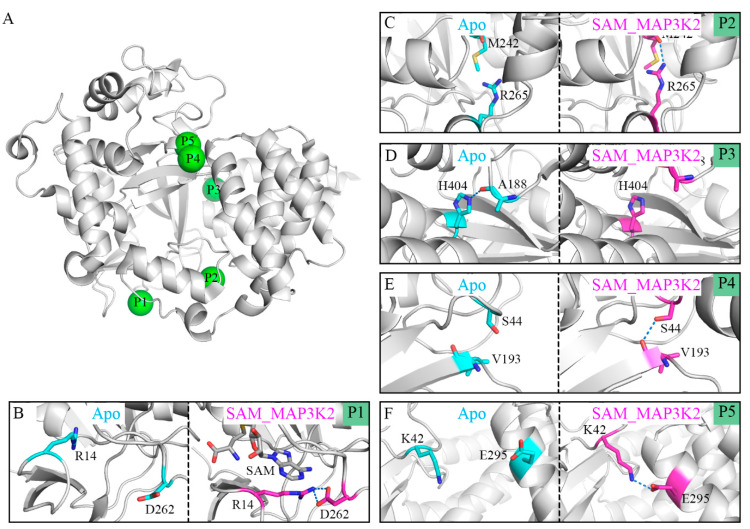
Distinct binding states of hydrogen bonds between the Apo and SAM_MAP3K2 systems. (**A**) P1–P5 positions labeled in the structure of SMYD3. (**B**–**F**) Binding states of hydrogen bonds at the P1–P5 positions in the Apo (cyan) and SAM_MAP3K2 (magenta) systems. Hydrogen bonds are depicted as dashed lines.

**Figure 5 ijms-22-07185-f005:**
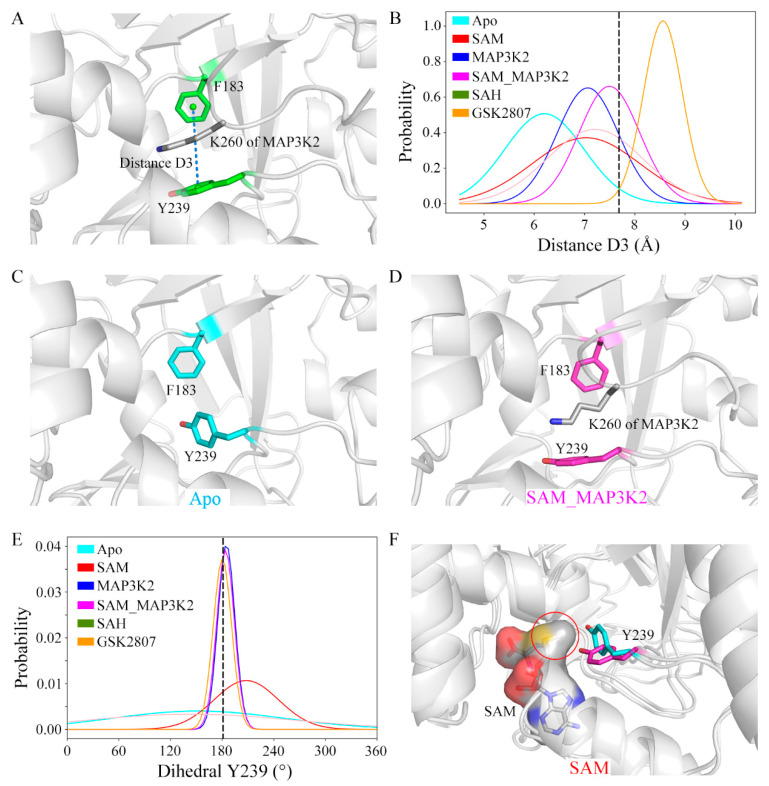
Structural details of the lysine-binding channel. (**A**) Definition of distance D3 in the structure of SMYD3. Residues in SMYD3 and the MAP3K2 peptide are shown as green and grey sticks, respectively. (**B**) Distributions of distance D3. The initial value in the crystal structure is labeled using a black dash. Conformational state of Y239 in the representative structures of the (**C**) Apo (cyan) and (**D**) SAM_MAP3K2 (magenta) systems. (**E**) Distributions of dihedral Y239. The initial value in the crystal structure is labeled using a black dash. (**F**) Two different conformational states of Y239 in the SAM system. Y239 in the initial and flipped conformational states are shown as magenta and cyan sticks, respectively. SAM is shown as grey sticks and with a grey surface.

**Figure 6 ijms-22-07185-f006:**
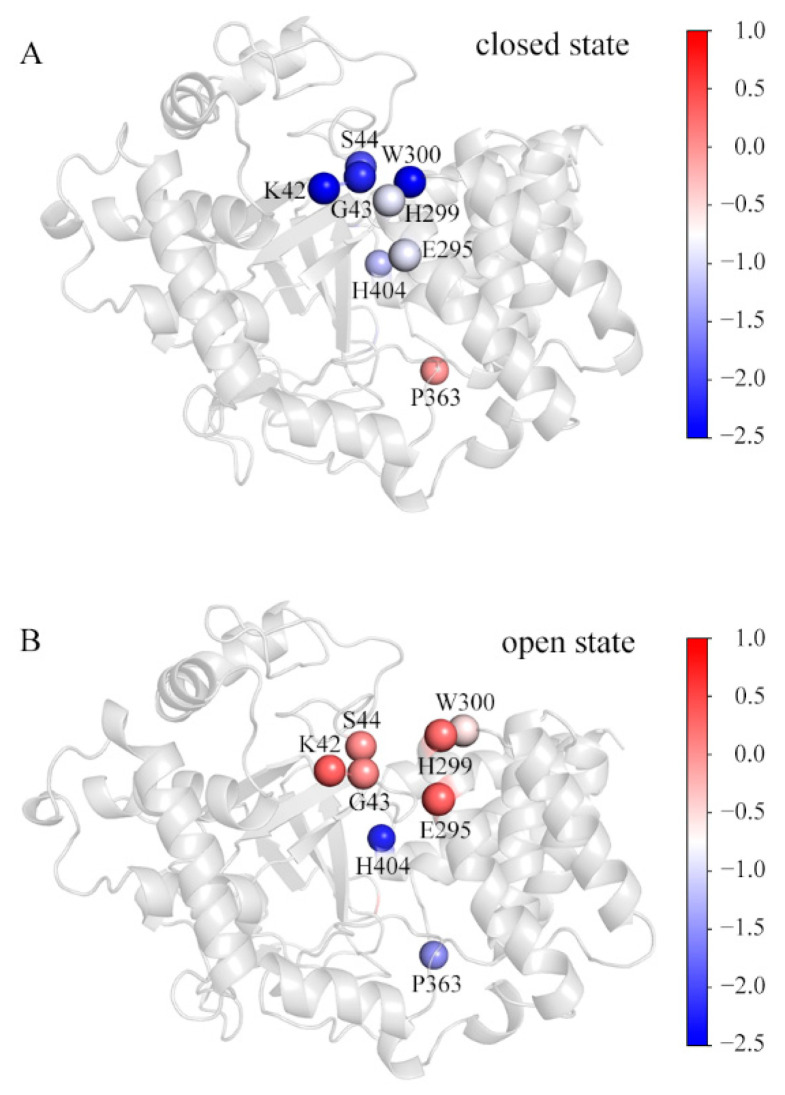
The decomposed binding free energy values of the SET and MYND domains toward the CTD. The representative structures in the (**A**) SAM_MAP3K2 and (**B**) Apo systems are colored by the decomposed binding free energy levels of the key residues. The color scale is shown on the right changing from red (high decomposed binding free energy) to blue (low decomposed binding free energy).

**Figure 7 ijms-22-07185-f007:**
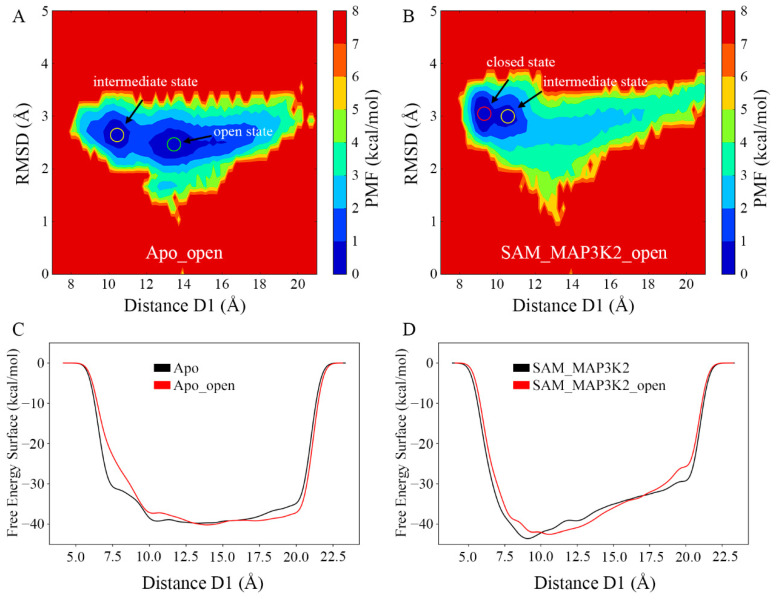
Conformational states of SMYD3 during MD simulations from the open conformation. PMF calculated for the distance D1 vs. the RMSD of SMYD3 for the (**A**) Apo_open and (**B**) SAM_MAP3K2_open systems. (**C**,**D**) Free energy surface associated with the conformational change of SMYD3 as a function of the distance D1 for the (**C**) Apo (black) and Apo_open (red), and (**D**) SAM_MAP3K2 (black) and SAM_MAP3K2_open (red) systems.

**Table 1 ijms-22-07185-t001:** Simulated models of SMYD3 with different components.

System Name	Components	Time Length (μs)	Box Size (Å^3^)	No. Water Molecules	No. Total Atoms
Apo	SMYD3	3 × 1	973,604.9	24,721	81,132
SAM	SMYD3, SAM	3 × 1	973,604.9	24,723	81,187
MAP3K2	SMYD3, MAP3K2 peptide	3 × 1	973,604.9	24,695	81,189
SAM_MAP3K2	SMYD3, SAM, MAP3K2 peptide	3 × 1	973,604.9	24,695	81,238
SAH	SMYD3, SAH	3 × 1	973,604.9	24,720	81,175
GSK2807	SMYD3, GSK2807	3 × 1	973,604.9	24,720	81,193
Apo_open	SMYD3	3 × 0.5	1,029,373.2	26,531	86,562
SAM_MAP3K2_open	SMYD3, SAM, MAP3K2 peptide	3 × 0.5	1,029,373.2	26,499	86,650

**Table 2 ijms-22-07185-t002:** Occupancies of hydrogen bonds at the P1–P5 positions during MD simulations.

Position	Hydrogen Bond	Occupancy					
		Apo	SAM	MAP3K2	SAM_MAP3K2	SAH	GSK2807
P1	R14-D262	1.76%	71.57%	5.58%	90.83%	93.61%	86.75%
P2	M242-R265	13.70%	66.65%	33.55%	48.38%	49.51%	61.28%
P3	A188-H404	63.17%	22.38%	52.33%	3.43%	31.34%	4.11%
P4	S44-V193	21.49%	53.94%	16.79%	87.23%	47.21%	82.77%
P5	K42-E295	5.73%	17.00%	37.63%	50.51%	11.47%	43.45%

**Table 3 ijms-22-07185-t003:** Binding free energies of the SET and MYND domains toward the CTD.

	Apo	SAM	MAP3K2	SAM_MAP3K2	SAH	GSK2807
Binding Free Energy ^a^	−17.72 ± 5.29	−15.77 ± 5.53	−18.90 ± 5.66	−32.41 ± 8.14	−16.93 ± 6.07	−30.91 ± 6.96

^a^ All binding free energies are in kcal/mol.

## Data Availability

Data related to this paper may be requested from the authors.

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
