# Peer review of "Mechanism of the Conformational Change of the Protein Methyltransferase SMYD3: A Molecular Dynamics Simulation Study"

_ijms, 2021, doi:10.3390/ijms22137185_

Round 1
Reviewer 1 Report
The manuscript entitled "Mechanism of the conformational change of the protein methyltransferase SMYD3: a molecular dynamics simulation study" presents a numerical study on the conformational changes of SMYD3. This work provides an insight into the behavior of SMYD3 by conducting comprehensive MD and metadynamics simulations. Before it can be accepted for publication, a few changes should be carried out.
It would be helpful if the authors provided a flowchart regarding the steps of their research in order to be more comprehensible. Moreover, Table 1 should be expanded in order to include more characteristics of each simulation case.
The authors chose to present the section "Materials and Methods" at the end of the manuscript. If it is not required by the format of the journal, then it could be more convenient to move this section before section 2. Furthermore, it is required that several more details are added to this section. The initial configuration of the MD models should be presented in appropriate figures and the number of atoms for each simulation should be also mentioned. The authors should present the results of the energy minimization and equilibration stages in order to prove that the selected number of steps for these stages were sufficient. Was the timestep value common to each simulation, e.g. 2 fs or not? The potential functions and parameter values used should be mentioned and justified.
Moreover, a few more details about the calculation of the PMF should be added as well as the formula for the calculation of RMSD.
Finally, the metadynamics simulation parameters should be presented in more details for each simulation and the validity of the simulations should be justified appropriately.
Reviewer 2 Report
The authors have investigated the SMYD3 protein by using MD simulation and extensive trajectories. The topic is generally interesting and the analysis performed standard, extensive and well-presented. Results are supported by the data. Methodology is well-described and literature also fine. There is always a question of how efficient this force-field really is. Since authors have presented a complete study of this system based on the choice of the force-field, I would like to recommend publication of this manuscript in ijms.
Author Response
We thank the reviewer’s positive comments of our work.
Round 2
Reviewer 1 Report
The authors performed the necessary modifications to their manuscript. Thus, it can be considered for publication. However, the authors should check again the numbering of their figures and whether all the figures mentioned in the text are named in the same way in the figure captions.